# Comparison of Different Dietary Fatty Acids Supplement on the Immune Response of Hybrid Grouper (*Epinephelus fuscoguttatus* × *Epinephelus lanceolatus*) Challenged with *Vibrio vulnificus*

**DOI:** 10.3390/biology11091288

**Published:** 2022-08-30

**Authors:** Maya Erna Natnan, Chen Fei Low, Chou Min Chong, Nur Iwani Nasuha Akiko Ahmad Daud, Ahmad Daud Om, Syarul Nataqain Baharum

**Affiliations:** 1Metabolomics Research Laboratory, Institute of Systems Biology (INBIOSIS), Universiti Kebangsaan Malaysia, Bangi 43600, Malaysia; 2Aquaculture Animal Health and Therapeutics Laboratory, Institute of Bioscience, Universiti Putra Malaysia, Serdang 43400, Malaysia; 3Department of Biosciences, Faculty of Science, Universiti Teknologi Malaysia, Johor Bahru 81310, Malaysia; 4Marine Fish Aquaculture Research Division, Fisheries Research Institute Tanjung Demong, Besut 22200, Malaysia

**Keywords:** oleic acid, fish immunity, immunology assay, aquafeed supplementation, vibriosis

## Abstract

**Simple Summary:**

Groupers are one of Asia’s most valuable marine fish. The strong market demand has driven the expansion of grouper farming. However, with the intense farming practices, the farm rearing system was exposed, thus increasing the risk of infectious diseases. The present study was conducted to compare the use of different fatty acid immunostimulants on the survival, growth, and immune response of hybrid grouper infected with *Vibrio vulnificus*. Our results showed that the oleic acid formulated diet gives the highest fish survival rate and growth rate compared to the other fatty acid formulated diets and control diet after six weeks of feeding trial and one week of the bacterial challenge. Moreover, fish supplemented with an oleic acid diet showed significantly increased immune responses after being infected with *V. vulnificus*.

**Abstract:**

Aquaculture has been expanding in Malaysia due to the increased demand for fish products. In addition, aquaculture faces challenges in maintaining feed suitability in support of the global growth of fish production. Therefore, improvements in diet formulation are necessary to achieve the optimal requirements and attain a desirable growth efficiency and health performance in fish. Seven weeks of study were conducted to compare the equal amounts of different fatty acids (2%) (oleic acid, stearic acid, palmitic acid, and behenic acid) on the survival, the growth, and the immune response of hybrid grouper (*Epinephelus fuscoguttatus* × *Epinephelus lanceolatus*) against *V. vulnificus*. After six weeks of the feeding trial, fish were challenged with *V. vulnificus* for 30 min before continuing on the same feeding regime for the next seven days (post-bacterial challenge). Fish supplemented with dietary oleic acid showed significantly (*p* < 0.05) enhanced immune responses, i.e., lysozyme, respiratory burst, and phagocytic activities compared to the control diet group for both pre-and post-bacterial challenges. Following the *Vibrio* challenge, no significant effects of supplemented fatty acid diets on survival rate were observed, although dietary oleic acid demonstrated the highest 63.3% survival rate compared to only 43.3% of the control diet group. In addition, there were no significant effects (*p* > 0.05) on specific growth rate (SGR), white blood cell (WBC), and red blood cell (RBC) counts among all experimental diets. The results from this study suggest that among the tested dietary fatty acids, the oleic acid diet showed promising results in the form of elevated immune responses and increased disease resistance of the hybrid grouper fingerlings challenged with *V. vulnificus.*

## 1. Introduction

Groupers are among the popular marine finfish cultured and traded in many Asian countries, including Malaysia [1]. However, due to the intensive farming techniques, they are at a high risk of exposure to infectious diseases. Vibriosis is the most common bacterial disease caused by the infection of *Vibrio* species. The rapid growth and the increased intensity of the mariculture activity caused an expansion of *Vibrio* sp. infections, which led to massive mortality of the cultured grouper and economic losses to the aquaculture sector in the country [2]. The symptoms of vibriosis include fin rot, swollen lesions on the skin, gill necrosis, lethargy, ulcers, darkened skin, and abnormal swimming pattern [3,4].

In the current aquaculture practices, antibiotics such as streptomycin, chloramphenicol, oxytetracycline, and sulfonamide are used to prevent and treat bacterial infections. However, several studies have suggested that these antibiotics should be avoided in fish culture. For example, the rampant use of these antibiotics has led to the emergence of antibiotic-resistant bacteria strains that caused the reduction of antibiotics efficacy [5]. Moreover, the extensive use of antibiotics also causes the accumulation of antibiotic residues in aquaculture products, which is associated with several food safety issues due to the contamination of these chemical agents [6].In other studies, some antibiotics have strong immunosuppressive effects on fish. For example, the use of oxytetracycline antibiotic not only causes oxidative stress in rainbow trout (*Oncorhynchus mykiss)* but also suppresses the fish’s immune responses, including lysozyme, superoxide dismutase (SOD), complement (ACH50), and glutathione peroxidase (GPx) activities [7,8]. Meanwhile, for florfenicol, the antibiotic not only inhibited the growth performance but also caused oxidative stress in European seabass (*Dicentrarchus labrax*), leading to lipid peroxidation of cell membranes, protein denaturation, and cellular damage [9]. Since there are some limitations to the use of antibiotics in aquaculture, a search for new alternatives that would provide good nutrition to the fish culture while ensuring food safety, quality, and environmental sustainability is necessary. The development of fish feed formulation with the use of additives such as amino acids, organic acids, and fatty acids as immunostimulants has given a new strategy for controlling infectious diseases in fish culture [10].

The innate immune response is a non-specific primary line of defense against pathogen invasion before inflammatory cells are employed to elicit non-specific protection and subsequently activate the adaptive immune response [11]. Upon exposure to pathogens, recognition of pathogen-associated molecular patterns (PAMPs) by the toll-like receptors (TLRs) stimulates the inflammatory response, which subsequently activates innate immunity [12].

Fatty acids are one of the important components in the fish diet that are utilized not only for growth, energy source, and structural cell membranes but also for enhancing the immune responses in fish [13,14]. Polyunsaturated fatty acids (PUFAs), for example, omega-3 PUFAs, are known to regulate the function of the immune response. Still, less attention has been paid to the effects of monounsaturated fatty acids (MUFAs) on the immune system [15]. The most common MUFA in daily nutrition is oleic acid (omega 9). Previous reports mentioned oleic acid as having health benefits because it can modulate signaling pathways in inflammatory responses, cell differentiation, and proliferation of immune cells [16,17]. Some studies reported that oleic acid is involved in different physiological functions of marine animals, including immunity and cell adhesion [18,19].In another study, oleic acid has a beneficial role in preventing excessive production of reactive oxygen species (ROS) and induces fatty acid oxidation during infection to prevent multiple organ failure and death. For example, oleic acid supplementation increased the survival rate and prevented liver and kidney injury of mice during sepsis [20]. Moreover, a study by Majdoubi et al. [21] demonstrated the composition of oleic acid and palmitic acid with other fatty acids such as docosahexaenoic acid (DHA) and eicosapentaenoic acid (EPA) showed improvement in the reproduction activity of the silver carp, *Hypophthalmichthys molitrix*. 

Based on our previous study, oleic acid, palmitic acid, stearic acid, behenic acid, palmitoleic acid, 8,11-eicosadienoic acid, and cis-erucic acid were highly abundant in surviving brown-marbled grouper, *Epinephelus fuscoguttatus* fingerlings that were challenged with *V. vulnificus* [22]. Therefore, in this study, we aimed to examine the effect of these fatty acid-based diet formulations, namely oleic acid, stearic acid, palmitic acid, and behenic acid, as potential immunostimulants. The survival rate, growth performance, and several innate immune parameters such as lysozyme activity, respiratory burst activity, and phagocytic activity of hybrid grouper fingerlings (*E. fuscoguttatus* × *E. lanceolatus*) challenged with *V. vulnificus* were monitored and evaluated. 

## 2. Materials and Methods

### 2.1. Experimental Diet Preparation 

A total of five diets were formulated to contain an equal amount (2%) of oleic acid, stearic acid, behenic acid, palmitic acid, and α-cellulose in the control diet, respectively. The basic diet formulation was prepared according to Ebrahimi et al. [23] with slight modifications. Fish meal and soybean meal were added as the main protein sources, while vegetable oil was added as the main lipid source. A study by Faudzi et al. [24] found that the concentration of soybean meal for hybrid grouper can reach up to 50% of the total feed formulation without significantly affecting the growth or body condition of the hybrid grouper. In addition, the increasing level of soybean meal content in the feed to 60% did not significantly affect the grouper feed intake [24,25]. In another study, soybean meal had been proven to partially replace fish meal without any adverse effect on the fish growth and the flesh meat quality [26]. In this experiment, cellulose was used to replace fatty acids in the control group because cellulose does not have any nutritional value for fish and is often used as a source of digestible carbohydrates in fish feed [27]. 

Ingredients (Table 1) were mixed thoroughly for 30 min, followed by adding vegetable oil and distilled water. The ingredients were then mixed for another 30 min. The mash was then pelleted through a commercial mincer with a 3 mm diameter die hole. The resultant strands were broken into small pellets (3 mm × 5 mm) and oven-dried at 45 °C overnight. The small particles were sieved before the pellets were kept in air-tight containers and stored at −20 °C prior to use. The proximate composition (Table 1) and the fatty acid profile (Table 2) of the experimental diets were measured according to the Association of Official Analytical Chemists (AOAC) 20th Edition.

### 2.2. Bacteria Culture

*V. vulnificus* was primarily obtained from infected liver, spleen, and kidney via fish dissection [22,28]. The bacterial strain was identified using a molecular approach. Samples from infected organs were aseptically streaked on thiosulphate citrate bile salt sucrose (TCBS) agar for *Vibrio* isolation. DNA extraction and polymerase chain reaction (PCR) procedure were performed using the 16S rDNA primers for *Vibrio*, VvForward: 5′-GTG GTA GTG TTA ATA GCA CT-3′ and VvReverse: 5′-GCT CAC TTT CGC AAG TTG GCC-3′. PCR products were then purified before being sent for sequencing. The sequence results were compared using The Basic Local Alignment Search Tool (BLASTn) search to find the regions of local similarity between nucleotide sequences in the NCBI GenBank database. The bacteria culture was revived from the glycerol stock on TCBS agar plates with the addition of 1.2% NaCl at 30 °C for 18–24 h. The inoculated bacteria culture was subcultured in tryptic soy broth (TSB) with the addition of 1.2% NaCl before being used for bacteria challenges.

### 2.3. Fish and Rearing Condition

A total of 250 hybrid grouper fingerlings of *E. fuscoguttatus* (female) × *E. lanceolatus* (male) were obtained from the local hatchery farm of Pantai Dasar Sabak, Kota Bahru, Kelantan. The groupers were reared and acclimatized at Hatchery Unit, Institute of Bioscience, Universiti Putra Malaysia, for one week in a 1000 L fiberglass tank filled with filtered seawater and kept aerated. The mean (±SD) grouper body weight used in the study was approximately 11.5 ± 0.5 g, and the length was about 3.0–4.0 inches.

### 2.4. Feeding Experimental Design 

After the initial acclimation period, a total of 225 healthy groupers were randomly distributed equally into five diet groups (*n* = 45 fish/diet group); each diet group was represented in three replicates (15 fish/replicate). The groupers were further acclimatized for five days while fed with the prepared control diet. After five days, the groupers were weighed before being fed with their respective diets containing oleic acid, palmitic acid, stearic acid, behenic acid, and control (feed without fatty acid supplement) twice daily at 9.00 am and 4.00 pm for six weeks (Appendix A). Based on previous studies, the six-week duration of the feeding trial is enough to develop protection within the fish’s immune response [29,30]. The feeding regime for the grouper was 4% body weight per day. The uneaten feed was removed after each feeding. Each aquarium was aerated and equipped with a water filter pump system throughout the experiment. Water quality was maintained by exchanging approximately 50% of the volume with clean seawater every two days. The water quality parameters were maintained at temperature 28.0 ± 1 °C, salinity at 30 ± 1 ppt and pH at 8.0 ± 1. At the end of the six-week experiment, the groupers’ final weights (g) were measured for specific growth rates (SGR) and feed conversion ratio (FCR). The SGR and FCR were calculated using the following equations:SGR (%day^−1^) = [ln (final weight) − ln (initial weight)]/T × 100
where, T = time in days:FCR = total weight of diet fed (g)/weight gain (g)

### 2.5. Sample Collection before the Bacterial Challenge 

A sample collection before the bacterial challenge was performed after six weeks of the feeding trial experiment without exposing the fish to *Vibrio*. Three fish from each tank were starved for 24 h before being anesthetized using MS-222 (50 mg/L). The peripheral blood and spleen were then sampled prior to immunology assays. The blood and spleen were pooled from three fish in the same tank to ensure the minimum required volume for immunology assay experiments (Figure 1). Blood samples subjected to respiratory burst activity assay were collected using a syringe with a 26-gauge needle rinsed with an ethylenediaminetetraacetic acid (EDTA) anticoagulant. For serum, a syringe without anticoagulant was used to draw the blood. The blood samples were then allowed to clot at room temperature before serum was collected by centrifugation at 5000 rpm for 10 min. The serum samples were stored at −20 °C prior to the lysozyme activity assay. The spleens were dissected from the groupers for phagocytosis activity assay. All assays were done as described in Section 2.8.1, Section 2.8.2, Section 2.8.3 and Section 2.8.4.

### 2.6. Fingerlings Challenged with LD50 of V. vulnificus

All methods used in the experiments followed the appropriate guidelines and regulations approved by UKM Animal Ethical Committee (UKMAEC) (Rujukan: IBC/Ack/2/2019). Briefly, after six weeks of being fed with five different fatty acid formulated diets comprising oleic acid, palmitic acid, stearic acid, behenic acid, and control (feed without fatty acid supplement), groupers were challenged with *Vibrio, V. vulnificus* grown in TSB with 1.2% NaCl for 18–24 h at 30 °C. The culture was adjusted to 5.4 × 10^7^ CFU/mL for the bacterial challenge test. Ten fish from each aquarium were randomly selected for the bacterial challenge test. The groupers were placed in their respective glass aquariums and challenged with *V. vulnificus* at the lethal dose (LD_50_) of 5.4 × 10^7^ CFU/mL by immersion for 30 min. The fish were then transferred into the new respective aquariums, and the feeding regime resumed for another seven days. The mortality rate within the seven days post-bacterial challenge was recorded. Blood and organs were sampled from the survived fish for subsequent analysis.

### 2.7. Sample Collection for Post-Bacterial Challenge

After seven days of post-bacterial challenge, three surviving groupers from each tank were starved for 24 h before being anesthetized, and the peripheral blood and spleen were sampled. The blood and spleen were pooled from three fish in the same tank to ensure the minimum required volume for immunology assay experiments (Figure 1). Blood samples subjected to cell counts and respiratory burst activity were collected using a syringe with a 26-gauge needle rinsed with an ethylenediaminetetraacetic acid (EDTA) anticoagulant. For serum, a syringe without anticoagulant was used to draw the blood. The blood samples were then allowed to clot at room temperature before serum was collected by centrifugation at 5000 rpm for 10 min. The serum samples were stored at −20 °C prior to the lysozyme activity assay. The spleens were dissected from the groupers for phagocytosis activity assay. All assays were done as described in Section 2.8.1, Section 2.8.2, Section 2.8.3 and Section 2.8.4.

### 2.8. Immunology Assay

#### 2.8.1. White Blood Cell (WBC) and Red Blood Cell (RBC) Counts

White blood cell count was determined in a dilution of 1:20 of the blood sample in 2% acetic acid, while RBC count was determined in 1:200 of the blood sample in 0.85% saline solution. The cells were counted using Neubauer haemacytometer chamber (Hirschmann, Laborgeräte GmbH & Co., Eberstadt, Germany) under light microscope. Triplicate samples were examined as described by Bakhshi et al. [31].

#### 2.8.2. Lysozyme Activity Assay

Serum lysozyme activity assay was performed according to Yeh et al. [32] and Bakhshi et al. [31] with slight modifications. A volume of 10 µL serum was mixed with 200 µL of *Micrococcus lysodeikticus* (cells purchased from Sigma-Aldrich, USA) suspension at the concentration of 0.2 mg/mL in 0.05 M sodium phosphate buffer (pH 6.2) on ice. The absorbance was recorded at 570 nm immediately (A1) using a microplate reader. The mixture was then incubated at 37 °C for 30 min. The reaction was stopped by incubation on ice prior to the second recording of the absorbance value (A2). One unit of lysozyme activity (U) is defined as the amount of enzyme producing a decrease in absorbance of 0.001/min mL/serums. The lysozyme activity was calculated according to the formulation: U = (A1 − A2)/A1. All samples were analyzed in triplicate, and the results were shown as an average.

#### 2.8.3. Respiratory Burst Activity Assay

Leukocyte respiratory burst activity was assayed using the collected blood samples according to Biller-Takahashi et al. [33] with modifications. Then, 50 μL of blood with anticoagulant was added to 50 μL of 0.2% nitroblue tetrazolium (NBT) solution (Sigma-Aldrich, USA) before it was homogenized and incubated for 30 min at 25 °C. The NBT solution was prepared in phosphate-buffered saline (PBS) in 1 L of distilled water), with adjusted pH of 7.4. After 30 min incubation, the sample was homogenized, and 50 μL of the sample was added to 1 mL of N, N-Dimethylformamide (DMF) (Sigma-Aldrich, St. Louis, Missouri, USA). The homogenized sample was then centrifuged at 3000× *g* for 5 min. A spectrophotometer at 540 nm was used to determine the supernatant’s optical density (OD). The blank consisted of the same components and steps, except blood was exchanged with distilled water. All samples were analyzed in triplicate, and the results were shown as an average.

#### 2.8.4. Phagocytic Activity Assay

Phagocytic activity was examined according to the method described by Romano et al. [34] and Cheng et al. [35] with slight modifications. Briefly, isolated grouper spleens were passed through a 40 µm cell strainer to obtain a single cell suspension. The single-cell suspension was then resuspended in 1 mL PBS and added with 500 µL of 34/51% (*v/v*) Percoll (Sigma-Aldrich, USA) prior to centrifugation at 1800 rpm for 5 min at room temperature. The interface obtained was then added with 500 µL of freeze medium as cryoprotectant agents for cell suspension before the samples were kept at −80 °C prior to use. The Freeze medium was prepared according to the American Type Culture Collection (ATCC) Animal Cell Culture Guide [36]. The splenic leukocytes were thawed and resuspended into 4–5 mL of PBS, then centrifuged at 1800 rpm for 5 min. The supernatant was removed, and the cells were resuspended again into 1 mL PBS. One part of 0.4% trypan blue was mixed with one part of the cell suspension. The mixture was then incubated for 1–3 min at room temperature, followed by microscopic examination using a Neubauer hemocytometer slide. The cells were counted and adjusted to a cell density of 5 × 10^6^ cells/mL using PBS. A volume of 50 µL spleen leucocyte suspension (5 × 10^6^ cells/mL) was loaded onto the glass slide and adhered in a moist incubation chamber for 20 min at 25 °C. Latex beads (polystyrene suspension; Sigma, USA) were added to the monolayer and incubated again for 30 min at 25 °C. After incubation, the slides were rinsed with saline and fixed with methanol for 5 min, then stained with Giemsa solution (Sigma, USA) for 15 min. The phagocytic activity was defined by the phagocytic index (PI) [37]. The number of beads ingested per phagocyte and the percentage of phagocytes ingesting beads were calculated by enumerating 100 phagocytes under microscopic examination. All samples were analyzed in triplicate, and the results were shown as an average.

### 2.9. Statistical Analysis

All results are presented as mean values of each group ± standard error (SE). All statistical analyses were performed using the statistical software program SPSS. All data were subjected to Shapiro–Wilk’s and Levene’s tests to test the normality distribution and homogeneity of variances. One-way ANOVA followed by Turkey’s post hoc test was performed on normally distributed data, namely SGR, FCR, and weight gain, to test the significant differences between the experimental groups. For data that were not normally distributed, such as survival rate, blood cell counts, lysozyme assay, respiratory burst assay, and phagocytic assay, a non-parametric Kruskal–Wallis’s test was performed to test the general differences between the experimental groups. Kruskal–Wallis’s post-hoc test was then used to compare the mean values between individual treatments. Immunology assay data from the pre-challenge and post-challenge were then subjected to a non-parametric Friedman’s 2-way ANOVA analysis in the SPSS to test the significant differences between pre-and post-bacterial challenges among each diet group. 

## 3. Results 

### 3.1. Growth Performance and Feeding Efficiency

After six weeks of treatment, the group of groupers that were fed with dietary oleic acid supplementation exhibited significant differences (*p* < 0.05) in the percentage of weight gain and FCR of 34.1% and 2.1, as compared to the control of 26.0% and 3.1, respectively (Table 3). Meanwhile, groupers fed with other fatty acid supplementations showed no significant effect on the parameters (Table 2) compared to the control supplementation. In addition, no significant variations in SGR (*p* > 0.05) were recorded among the five group diets.

### 3.2. Survival and Hematological Parameters in Post-Challenge

First mortality was recorded on day 3 post-challenge. The infected grouper exhibited skin lesions, pale gills, and discoloration (Figure 2b). Based on BLASTn search, the sequences of the isolated bacteria showed high homology (99%) with *V. vulnificus* (Appendix A). The grouper did not have a significant difference in survival rate (*p* > 0.05) after seven days of the post-bacterial challenge. In Table 4, it was observed that groupers fed with dietary oleic acid showed a higher percentage of survival rate (63.3%) compared to the control (43.3%) and other dietary fatty acids, including stearic acid, palmitic acid, and behenic acid with 53.3%, 53.3%, and 50.0%, respectively (Table 4). Moreover, no significant difference (*p* > 0.05) was recorded for red blood cell (RBC) count and white blood cell (WBC) count among all feeding treatments after seven days of the post-bacterial challenge.

### 3.3. Lysozyme Activity, Respiratory Burst Activity, and Phagocytic Activity Assays 

In all treatment groups, immunology assays for lysozyme, respiratory burst, and phagocytic assays showed higher activity in the post-bacterial challenge than in the pre-bacterial challenge conditions (Figure 3). Kruskal–Wallis’s analysis demonstrated that dietary fatty acids had a significant effect on lysozyme, respiratory burst, and phagocytic activity levels, as presented by the significantly higher level of lysozyme, respiratory burst, and phagocytic activities in groupers fed with dietary fatty acid formulations compared to the control diet group (*p* < 0.05). 

As reported in Figure 3A, lysozyme activity was significantly (*p* < 0.05) higher in groupers fed with dietary oleic acid and dietary stearic acid formulations compared to the groupers fed with a control diet in both pre- and post-bacterial challenge conditions. Meanwhile, no significant difference (*p* > 0.05) was recorded between groupers fed with control, behenic acid, and palmitic acid diet formulations in pre- and post-bacterial challenge conditions. (Figure 3A). 

Respiratory burst activity in the pre-bacterial challenge and post-bacterial challenge groups are presented in Figure 3B. Compared to the control diet group, Kruskal–Willis post hoc analysis demonstrated that groupers fed dietary oleic acid and stearic acid formulations were highly significant (*p* < 0.05) compared to the control formulation diet in the post-bacterial challenge. Meanwhile, in the post-bacterial challenge, no significant difference could be observed for groupers fed with the control diet compared to those fed with dietary behenic acid and palmitic acid. For the pre-bacterial challenge, no significant difference in leukocyte respiratory burst activity was observed among all fatty acid diets and the control diet group (*p* > 0.05) (Figure 3B). 

As shown in Figure 3C, the phagocytic activity in post-bacterial challenge revealed that groupers fed with oleic acid supplementation exhibited a significantly (*p* < 0.05) higher phagocytosis rate than the groupers fed with the control diet. No significant effects (*p* > 0.05) on phagocytic activity levels were observed among control, stearic acid, behenic acid, and palmitic acid diets in response to the post-bacterial challenge (Figure 3C). Meanwhile, in the pre-challenge groups, dietary treatments of oleic acid and stearic acid had a significant effect on phagocytic activity levels compared to the control diet (*p* < 0.05). In contrast, no significant effects were observed among groupers fed with palmitic acid, behenic acid, and control diets in the pre-bacterial challenge conditions (*p* > 0.05). 

Additionally, as reported in Figure 3A–C, in response to the post-bacterial challenge with *V. vulnificus*, significant increases in groupers’ immunity response, including lysozyme, respiratory burst, and phagocytic activities, were shown on all dietary fatty acids for the post-bacterial challenge when compared with the pre-challenge groups (Friedman’s test, *p* < 0.05).

## 4. Discussion

This study used fatty acid compounds as an additional supplement in the fish meal diet. Fatty acids are known as essential nutrients, like others, such as amino acids, vitamins, and minerals. Additionally, other macronutrients, for example, protein, carbohydrate, and lipid are necessary to meet the specific nutrient requirements of fish at different life stages [38]. Emerging technology such as metabolomics is crucial in obtaining new knowledge about the fish immune system and its contribution to the development and prevention of aquaculture diseases [39]. Based on our previous metabolomics analysis using the gas chromatography-mass spectrometer (GCMS) approach, several metabolites, such as oleic acid, palmitoleic acid, palmitic acid, 8,11-eicosadienoic acid, behenic acid, stearic acid, and cis-erucic acid from the fatty acid group were found highly abundant in survived *E. fuscoguttatus* during the experimental infection by *V. vulnificus* [22,28]. In our current study, only four fatty acids, namely, oleic acid, palmitic acid, behenic acid, and stearic acid, were selected to be used in the fish diet formulation and investigated their potential effects on improving innate immune response and resistance of grouper against *Vibrio* infection. The four fatty acids chosen for the fish formulations are not only largely dependent on their high metabolite abundant in the grouper challenged with *Vibrio* but also because of their availability and cost-effectiveness in the market as the low cost of fish feed ingredients will reduce the feed costs and benefit the aquaculture producers. The results indicated that the supplementation of fatty acids in fish diets improved the innate immune responses of the hybrid grouper. This was in line with the previous report, which mentioned that fatty acids could manipulate immune responses by initiating various processes, including lipid peroxide formulation, membrane fluidity, gene regulation, and eicosanoid production [40,41,42]. 

In the present study, the addition of oleic acid in the formulated diet significantly affected the innate immune responses of hybrid grouper fingerlings. The current result agrees with the previous findings, in which oleic acid from olive oil increased the survival rate and induced inflammation of large yellow croaker, *Larimichthys crocea* [43]. This study examined red and white blood cell counts after one week of the *Vibrio* challenge. White blood cells consist of lymphocytes that function to produce antibodies against pathogen infection. At the same time, macrophages, monocytes, and neutrophils are specialized phagocytes that eliminate foreign microorganisms through phagocytosis [44,45]. Hematological parameters are important diagnostic tools for fish health monitoring as they are a reliable indicator of the responses of fish to environmental stress or diseases via non-lethal procedures [46,47]. In the current study, the WBC count was attributable to the peaking number of granulocytes indicating stress. The increased WBC count indicated the defense mechanism of the fish to stress and bacterial infection [48]. No significant difference between all groups indicated that the formulated diets were not stressful to the fish.

Fatty acids are important dietary sources of growth, energy, and major components of cell membranes, and precursors to signaling molecules [49] within the fish body. In the current study, although fatty acid supplementation positively improved the immune response and survival rate of grouper, there is limited information and studies conducted by other researchers about the effects of these fatty acids on the growth rate, immune response, and survival of the fish species. In the current study, the dietary ratio of 2% of fatty acid was chosen based on Ebrahimi et al. [23] study, where this ratio used for dietary feed preparation showed optimum results in inducing growth and immune response for red hybrid tilapia (*Oreochromis* sp.). Similar levels of the diet containing α-linolenic (LN) and linoleic acids (LA) were beneficial to weight gain and non-specific cellular immune response of juvenile Malabar grouper, *Epinephelus malabaricus* [50]. They revealed that a diet containing 2% of LN and LA showed significantly higher phagocytic and respiratory burst activities compared with the other treatments with lower levels of pure fatty acids of LN or LA. On the contrary, in other studies, the intake of more than 2% of the total prepared feed formulation of fatty acids, such as n-3 or n-6 PUFAs, may suppress the immune responses of several fishes, including juvenile golden pompano, *Trachinotus ovatus* [51] and Atlantic salmon, *Salmo salar* [52]. In the present study, grouper fed with a diet containing 2% oleic acid showed significantly higher immune responses among the treatment groups in pre- and post-bacterial challenge conditions. The finding was comparable to those reported in a previous study by Nayak et al. [53], where the long-chain polyunsaturated fatty acids from the omega-6 class, known as arachidonic acid (ARA) and dihomo-Ƴ- linolenic acid (DGLA) demonstrated the potential of ARA- and DGLA-derived dietary supplements in improving immune response and resistance to bacterial infection of *Streptococcus iniae* in zebrafish. The ARA dietary supplement has also shown positive effects on growth, survival, immune responses, and lipid metabolism in other fish, such as Malabar red snapper, *Lutjanus malabaricus* [54], European seabass, *D. labrax* [55], and Japanese eel, *Anguilla japonica* [56]. In another study, the supplementation with ARA increased lysozyme, phagocytosis, and superoxide dismutase activities in Malabar red snapper (*L. malabaricus*) fingerlings [54]. Similar results were also found in juvenile turbot, *Scophthalmus maximus* L., where the composition of an ARA-enriched diet triggered the metabolic production of eicosanoid, which promoted the anti-inflammatory cytokine and reduced the pro-inflammatory cytokine responses [57].

Lysozyme is an enzyme of the innate immune response [58] that acts as the main effector molecule in anti-bacterial and anti-inflammatory activities during infection [59]. Meanwhile, phagocytosis and respiratory burst activities are macrophages, neutrophils, and eosinophils that consume oxygen to release various reactive oxygen species (ROSs), including superoxide anion radical (O2^–^), hydrogen peroxide (H_2_O_2_), and hydroxyl radical (OH^–^) [60]. Our findings revealed that fish fed with oleic acid exhibited increased lysozyme, respiratory burst, and phagocytosis activities. In this research, the highest lysozyme activity, respiratory burst activity, and phagocytic activity can be observed in both pre-, and post-bacterial challenges of grouper fed with the dietary oleic acid (Figure 3A–C). However, when comparing the immune activities between the pre-and post-bacterial challenge conditions, our research showed the immune response activities were higher in the post-bacterial challenge condition than in the pre-bacterial challenge condition (Figure 3A–C). This is in agreement with the enhanced respiratory burst activity, phagocytic activity and lysozyme activity in sutchi catfish, *Pangasianodon hypophthalmus* fed with fucoidan rich seaweed extract, where their immune response activities were higher in the post-bacterial challenge conditions than the pre-bacterial challenge condition [61]. The higher number of phagocytic cells and other immune response activities in the post-bacterial challenge condition could be related to the elimination of the invading microorganisms [62,63], which in our study was more persistent in the fish infected with *V. vulnificus* after seven days of the post-bacterial challenge. In another study, similar results were shown for pacu, *Piaractus mesopotamicus* fed with β-glucan, where the respiratory burst activity and lysozyme activity were higher after *A. hydrophila* infection than before the bacterial infection [54]. The increased respiratory burst activity in challenged fish indicates that phagocytes elevate the activation of the NADPH oxidase enzyme to produce ROSs that damage the cell and simultaneously destroy invasive pathogens [60]. The same principle can be applied in our study in which the increase of phagocyte activity in grouper also increased the amount of secreted lysozyme during pathogenic infection as in Asian seabass, *Lates calcarifer* against *Vibrio alginolyticus* infection [64].

Several publications have shown the potential role of vegetable oils containing oleic acid composition as an immunostimulant for inducing immune responses. Previous reports mentioned that oleic acid could modulate the signaling pathways in the inflammatory response, cell differentiation, and proliferation of immune cells [16,17]. A study by Sankian et al. [65] demonstrated that the survival and the growth performance of mandarin fish, *Siniperca scherzeri* was remarkably improved with the daily feed intake of soybean oil and linseed oil that contained high fatty acid content, such as oleic acid, linoleic acid, and α-linoleic acid. These results agree with a study where oleic acid played a role in activating the different pathways of immune-competent cells [66] and is responsible for modulating wound inflammation by inducing wound healing [17]. Similarly, a study by Li et al. [43] on large yellow croaker, *L. crocea,* fed with olive oil containing a high-level composition of the oleic acid compound showed an increment of pro-inflammatory gene expression and increased activity for other immune-related proteins and enzymes, such as tumor necrosis factor-α (TNFα), interleukin-1β (IL-1β) and cyclooxygenase-2 (COX-2). Moreover, another study revealed that oleic acid has antibacterial activity, which inhibits the growth of several pathogens. For example, treatment of bivalve, *Argopecten purpuratus* with the oleic acid isolated from bacteria reduces the bacterial load of *Vibrio parahaemolyticus* [67]. Meanwhile, the squid ink extract containing a high inclusion of oleic acid had been reported as an anti-*Vibrio* agent in the treatment of *Vibrio alginolyticus*-infected grouper juveniles [44]. 

Other fatty acid diets such as palmitic acid, stearic acid, and behenic acid showed satisfactory results in inducing the innate immune response due to bacterial infection. These findings agree with previous studies showing palmitic acid was used as an immunostimulant to reduce the mortality of zebrafish infected with spring viremia of carp virus (SVCV). Here, zebrafish larvae treated with palmitic acid activated the anti-viral mechanisms that inhibited the autophagic flux and increased the neutrophil proliferation and type 1 interferon (IFN) activity during the viral infection [68]. Meanwhile, in another study, zebrafish larvae treated with different proportions of palmitic acid showed an increment of ROS and nitrous oxide production to activate the pro-inflammatory cytokines [69]. As for stearic acid and behenic acid, although no specific study was found related to immune response, these fatty acids have been reported to show good antibacterial activity against several pathogens. For example, the novel stearic acid analogs similar to stearic acid isolated from the microalgae, *Spirulina platensis* showed antibacterial activity against pathogens, including *Staphylococcus aureus*, *Escherichia coli*, *Bacillus subtilis,* and *Pseudomonas aeruginosa* [70]. Meanwhile, for behenic acid, it was reported that the composition of behenic acid, palmitic acid, oleic acid, linoleic acid, linolenic acid, and myristic acid in the leaves of *Sesuvium portulacastrum L.* showed the potential of anti-microbial and anti-fungal agents against *B. subtilis*, *Aspergillus fumigatus,* and *Aspergillus niger* [71].

## 5. Conclusions

In conclusion, the use of oleic acid as an immunostimulant for juvenile grouper’s fish feed demonstrated significant positive effects on the enhancement of immune responses regarding lysozyme activity and respiratory burst activity as well as phagocytic activity in both pre- and post-bacteria challenge. The dietary oleic acid provided higher growth performance and survival of the hybrid grouper against *V. vulnificus* infection. Therefore, it can be suggested that feed manufacturers could use oleic acid in the aquaculture industry as a new nutritional strategy for minimizing economic losses due to infectious diseases. Further investigations on the feed pellet dissolution and the optimal supplementation for groupers are recommended to improve the fish feed for aquaculture use. 

## Figures and Tables

**Figure 1 biology-11-01288-f001:**
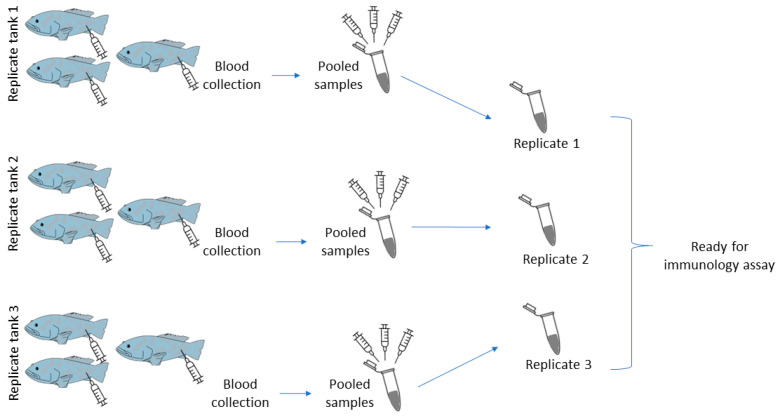
Diagram for blood sampling after feeding trial and after post-bacterial challenge. The same sampling procedure was carried out for spleen collection (pooled sample n = 3).

**Figure 2 biology-11-01288-f002:**
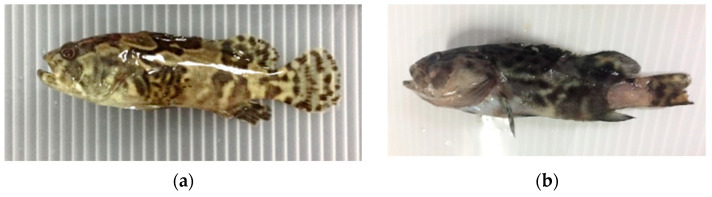
Picture of grouper with no bacteria challenge (**a**). Physical symptoms of grouper challenged with *V. vulnificus* (**b**).

**Figure 3 biology-11-01288-f003:**
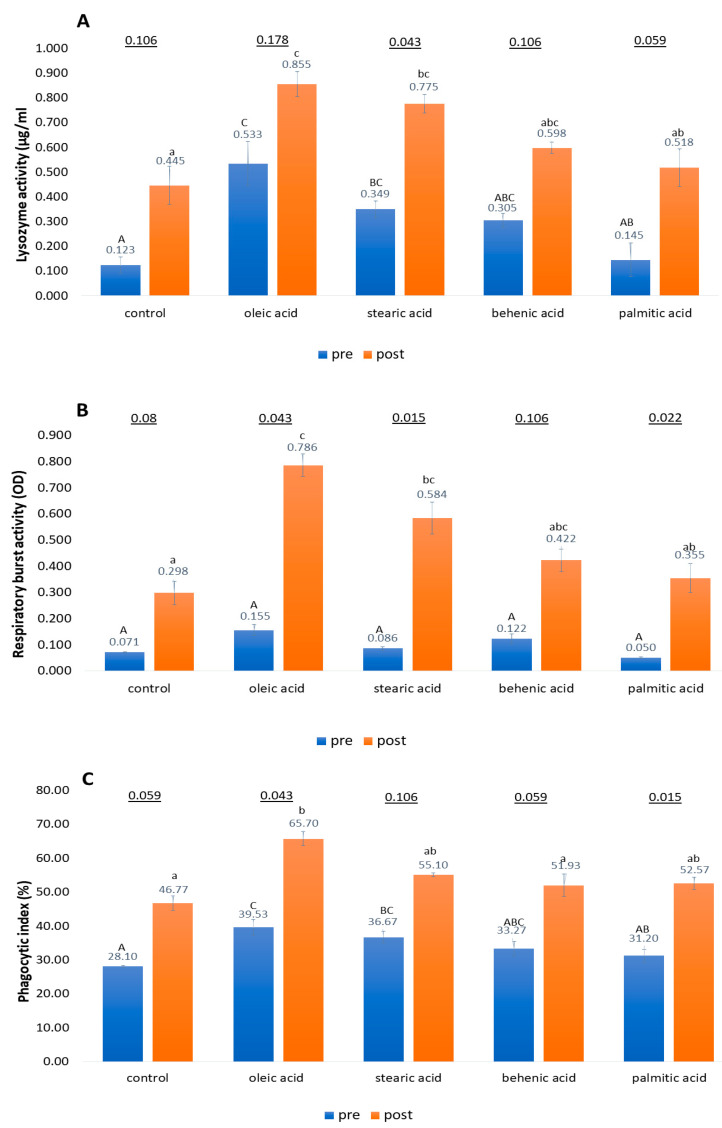
Lysozyme (**A**), respiratory burst (**B**), and phagocytic (**C**) activity changes after pre- and post-bacterial challenge of survived grouper fingerlings fed with control, oleic acid, stearic acid, behenic acid, and palmitic acid supplemented diets. The bar represents the mean (±SE) values of three replicates. *p*-values on the top of the bar denote significant differences between pre-and post-bacterial challenges among the same diet (Friedman’s test, *p* < 0.05). Different letters (A, B, C for the pre-bacterial challenge; a, b, c for the post-bacterial challenge) above in each bar indicate significant differences (*p* < 0.05) in immunology changes in groupers fed with five different diets (analyzed by Kruskal–Wallis’s post-hoc test for multiple comparison test).

**Table 1 biology-11-01288-t001:** Ingredient formulation (%) of the experimental diets with different types of fatty acid addition.

Experimental Diets					
Ingredients	Control	Oleic Acid	Stearic Acid	Behenic Acid	Palmitic Acid
Fish meal	11.5	11.5	11.5	11.5	11.5
Soybean meal	50.5	50.5	50.5	50.5	50.5
Vegetable oil	7.5	7.5	7.5	7.5	7.5
Corn flour	24.0	24.0	24.0	24.0	24.0
Vitamin mixture ^a^	2.0	2.0	2.0	2.0	2.0
Mineral mixture ^b^	2.0	2.0	2.0	2.0	2.0
Oleic acid ^c^	0.0	2.0	0.0	0.0	0.0
Stearic acid ^d^	0.0	0.0	2.0	0.0	0.0
Behenic acid ^e^	0.0	0.0	0.0	2.0	0.0
Palmitic acid ^f^	0.0	0.0	0.0	0.0	2.0
α-Cellulose ^g^	2.5	0.5	0.5	0.5	0.5
**Proximate composition ^h^**					
Total ash	6.9	6.7	6.8	6.7	6.7
Moisture	6.7	6.8	6.8	8.7	7.6
Protein	31.3	31.4	31.0	30.8	31.3
Lipid	9.4	11.4	11.3	11.1	11.2
Carbohydrate	45.7	43.7	44.1	42.7	43.2
Energy (kcl/100 g)	393	403	402	394	399

^a^ Vitamin mixture per kg: vitamin A, 5,000,000 IU; vitamin B2, 15,000 IU; vitamin D3, 1,000,000 IU; vitamin B6, 12,000 IU; vitamin B1, 15,000 mg; vitamin B12, 25 mg; vitamin C, 300,000 mg; vitamin E, 50,000 mg; vitamin K3, 5000 mg; biotin, 500 mg; folic acid, 2500 mg; pantothenic acid, 25,000 mg; inositol, 125,000 mg; niacin, 50,000 mg (Dexchem Industries Sdn. Bhd). ^b^ Mineral mixture per kg: KCl, 50,000 mg; NaCl, 60,000 mg; MgSO₄, 137,000 mg; Fe, 15,000 mg; Zn, 1894 mg; Cu, 785 mg; iodine, 150 mg; Mn, 800 mg; Co, 100 mg; Na, 20 mg; dicalcium phosphate, 723,251 mg; anti caking, 1000 mg (Dexchem Industries Sdn. Bhd, (Penang, Malaysia). ^c^ oleic acid (R&M Chemicals, UK 112801). ^d^ stearic acid (R&M Chemicals, UK 57114). ^e^ behenic acid (Alfa Aesar, Haverhill, MA, USA A12850). ^f^ palmitic acid (R&M Chemicals, UK 57103). ^g^ α-Cellulose (Sigma-Aldrich Co., St. Louis, Missouri, USA C8002). ^h^ Determined according to the Association of Official Analytical Chemists (AOAC) 20th Edition.

**Table 2 biology-11-01288-t002:** Fatty acid (mg/100 g on dry matter basis) profile of the formulated experimental diets.

	Control	Oleic Acid	Stearic Acid	Behenic Acid	Palmitic Acid
Saturated fatty acids					
C4:0	0.19	0.65	0.00	0.16	0.22
C8:0	0.58	0.81	0.61	0.66	0.61
C10:0	0.51	0.76	0.50	0.51	0.61
C11:0	0.54	1.00	0.59	0.64	0.64
C12:0	3.41	15.11	4.26	4.59	3.78
C13:0	0.00	1.06	0.20	0.20	0.31
C14:0	19.45	30.39	19.43	26.60	19.60
C15:0	5.55	7.61	5.50	6.50	5.67
C16:0	947.56	877.36	1311.84	1302.02	1245.07
C17:0	9.14	10.34	10.83	9.80	10.04
C18:0	204.02	221.62	257.93	263.62	250.52
C20:0	49.43	48.15	61.76	64.91	61.03
C22:0	21.27	23.25	29.58	38.31	27.21
C23:0	14.70	23.58	18.56	9.74	11.39
C24:0	32.13	83.99	39.74	53.53	34.80
Total	1308.47	1345.68	1761.34	1781.81	1671.49
Monounsaturated fatty acids					
C14:1	1.39	2.77	0.89	0.97	0.86
C15:1	0.97	1.69	0.95	1.20	1.11
C16:1	32.01	40.31	30.70	42.16	31.89
C17:1	7.67	13.04	7.96	7.80	8.61
C18:1n9 cis	3144.73	4536.41	3576.96	3548.18	3601.36
C20:1n9	46.22	57.29	51.91	55.10	52.15
C22:1n9	2.32	6.63	8.52	9.81	4.24
C24:1	13.83	135.28	27.23	48.09	18.57
Total	3249.15	4793.43	3705.12	3713.31	3718.78
Polyunsaturated fatty acids					
C18:2n6 cis	4633.96	5001.67	5601.85	5363.93	5570.41
C18:3n6	10.60	7.23	13.56	9.28	12.40
C18:3n3	116.47	134.44	133.04	130.87	142.08
C20:2	3.34	4.96	3.43	3.65	3.43
C20:3n3	8.41	16.99	8.27	9.22	7.82
C20:4n6	4.77	5.67	4.58	4.58	6.69
C20:5n3	15.11	23.45	15.61	21.17	17.45
C22:2	10.82	7.62	13.82	12.75	7.79
C22:6n3	38.89	58.86	39.36	49.43	41.67
Total	4842.38	5260.89	5833.54	5604.88	5809.73

Determined according to the Association of Official Analytical Chemists (AOAC) 20th Edition.

**Table 3 biology-11-01288-t003:** Growth performance and feed efficiency of grouper fingerlings at six-week post-feeding with five different dietary fatty acids.

	Control	Oleic Acid	Stearic Acid	Behenic Acid	Palmitic Acid
Initial weight (g)	11.37 ± 0.33 ^a^	12.03 ± 0.07 ^a^	11.30 ± 0.45 ^a^	11.82 ± 0.42 ^a^	11.73 ± 0.23 ^a^
Final weight (g)	14.32 ± 0.22 ^a^	16.13 ± 0.15 ^b^	14.64 ± 0.35 ^ab^	15.49 ± 0.12 ^bc^	14.96 ± 0.32 ^ac^
Weight gain (%)	26.00 ± 1.40 ^a^	34.10 ± 1.00 ^b^	29.60 ± 3.10 ^ab^	31.10 ± 1.00 ^ab^	27.50 ± 1.20 ^ab^
SGR (%/day)	0.55 ± 0.03 ^a^	0.70 ± 0.02 ^a^	0.62 ± 0.06 ^a^	0.64 ± 0.02 ^a^	0.58 ± 0.02 ^a^
FCR	3.1 ^a^	2.1 ^b^	2.8 ^ab^	2.3 ^ab^	2.9 ^a^

Different superscripts in the same row indicate a significant difference (*p* < 0.05) between different fatty acid diets. Values are represented by mean ± standard error (SE). (One-way ANOVA, followed by Turkey’s post hoc multiple comparison test).

**Table 4 biology-11-01288-t004:** Survival rate and blood cell counts of hybrid grouper fed with different fatty acid diets after 7-day of post-bacterial challenge.

	Control	Oleic Acid	Stearic Acid	Behenic Acid	Palmitic Acid
Total survival/initial fish used	13/30	19/30	16/30	15/30	16/30
Survival rate (%)	43.30 ± 5.77 ^a^	63.30 ± 8.82 ^a^	53.30 ± 3.33 ^a^	50.00 ± 5.77 ^a^	53.30 ± 3.33 ^a^
RBC count (×10^6^)/mm^3^	1.12 ± 0.07 ^a^	1.32 ± 0.08 ^a^	1.18 ± 0.01 ^a^	1.22 ± 0.06 ^a^	1.21 ± 0.03 ^a^
WBC count (×10^3^)/mm^3^	3.10 ± 0.10 ^a^	3.98 ± 0.18 ^a^	3.45 ± 0.25 ^a^	3.50 ± 0.20 ^a^	3.18 ± 0.13 ^a^

Different superscripts in the same row indicate a significant difference (*p* < 0.05) between different fatty acid diets. Values are averages ± SE. RBC; red blood cells; WBC, white blood cells. (non-parametric Kruskal–Wallis’s test followed by Kruskal–Wallis post-hoc multiple comparison test).

## Data Availability

Not applicable.

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
