# Peer review of "Comparison of Different Dietary Fatty Acids Supplement on the Immune Response of Hybrid Grouper (Epinephelus fuscoguttatus × Epinephelus lanceolatus) Challenged with Vibrio vulnificus"

_biology, 2022, doi:10.3390/biology11091288_

Round 1
Reviewer 1 Report
The manuscript examined the effect of different fatty acids supplementation in practical diets for hybrid group on immune responses and survival after challenge with Vibrio vulnificus. The topic is relevant and important to the development of feed supplements for marine fish. However, the rationale for the study, particularly the choice of fatty acids supplement tested, was not well justified. The analyses and the subsequent interpretation were good.
1. Please provide a justification why the four fatty acids were chosen to test as immunostimulant among the list of compounds identified from the previous study. Why compare an omega 9 unsaturated fatty acid supplement with three other saturated fatty acids? Considering there are also other omega 9 fatty acids in the list as well.
2. What vegetable oil (main lipid source) was used in the diet formulation? Have the essential fatty acids requirement of the fish being met with these diets?
3. Please explain why alpha-cellulose was used to balance the fatty acid supplemented formulation instead of other ingredient? With a controlled ration (4% body weight), fish in the control group would receive less calories than other groups. What was your rationale?
4. Please add proximate analysis results (protein, lipid, moisture), total calories, and fatty acid profile of the diets to confirm your formulation.
5. Immune response results are presented in both in the table and figure format. I do not think both needs to be included.
There are some minor problems with English.
Title: "Comparison of different fatty acids dietary .... " should be "Comparison of different dietary fatty acids". Also, with 2% addition, "supplement" might be a better term to include in the title.
Figure 1: All the treatment name (".... dietary feeding") - consider using "diet" or "feed" instead. Be careful with the use of the word feed/feeding/diet/ dietary as they are differences in usage
Reviewer 2 Report
The Authors of the reviewed manuscript titled "Comparison of different fatty acids dietary on the immune response of hybrid grouper (Epinephelus fuscoguttatus x Epinephelus lanceolatus) challenged with Vibrio vulnificus" have presented a mediocre paper, which describes relatively simple results obtained in a single-experiment study. While the content of this work is publishable, it is nothing of a revolutionary magnitude, as there have been plenty of more complicated analyses performed in the past which highlighted the immunostimulatory effects of fatty acids on different fish species. Nevertheless, data analysis within this paper needs to be improved, as the presented statistical calculations are not scientifically sound.
I have arranged my specific line to line commentary below, paragraph by paragraph. I also marked the same changes in the attached PDF file. Most of the points are in regard to necessary linguistic corrections, but, in terms of the English language, more stylistic improvements could be made by the Authors during revision.
Title: Change word order to "dietary fatty acids".
Simple Summary: Apart from the lacking first 2-3 sentences, the rest of this section seems to be quite fitting.
Lines 17-18: This sentence is of poor style and repeats words from the previous sentence ("high demand"). "Increase in culturing"? Please rephrase and/or restructure.
Lines 19-20: Again, it is barely understandable what is meant here. This sentence needs urgent rewriting.
Line 21: Correct to singular "acid".
Abstract: Like the simple summary, some linguistic corrections are required. Content-wise this section seems to be OK.
Lines 27-28: Delete "locally and globally".
Lines 28-29: This sentence needs rewriting from scratch.
Line 33: Delete the parenthesis "(pre-bacterial challenged)".
Line 34: Delete "one more". Also correct to present "challenge".
Line 35: Change word order to "dietary oleic acid". Also change "led to" to "showed".
Line 37: Correct to present "challenge".
Line 41: Correct to present "suggest".
Lines 41-42: Change word order to "dietary fatty acids". Also, replace "on" with "in the form of".
Line 43: Change to plural "fingerlings".
Keywords: I suggest to change the keyword "fish dietary" to maybe "aquafeed supplementation".
Introduction: This part is quite clear and includes all the necessary context of the study. I have outlined the necessary linguistic corrections which need to be made, but in terms of style, more improvements could be done here. Nevertheless, this is a decently written section.
Line 52: Correct "the expanded list" to "an expansion of".
Line 53: Delete "the" infront of "economic".
Line 54: Change to singular "fin rot".
Line 55: Change to plural "lesions".
Line 62: Change "of" to "with".
Line 63: Please rephrase this part: "showed some restrictions used in aquaculture", there are some mistakes here.
Line 64: Change "provides" to "would provide".
Line 66: Replace "from" with "with the use of".
Line 67: Delete "insight and".
Line 75: Correct "Fatty acid is" to plural "Fatty acids are". Also, change "used" to "are utilized".
Line 77: Correct "involve" to "is involved".
Line 84: Correct "contained with" to "containing".
Line 87: Correct to "surviving".
Line 88: Correct to plural "fingerlings".
Line 89: Correct to singular "acid".
Materials and Methods: This section, apart from a few omissions, was written truly comprehensively. Well done.
Line 96: Change to singular "diet".
Line 102: Delete "be".
Line 103: Correct to "affecting".
Line 110: Correct to singular "acid".
Lines 114-119: The citation "[16]" suggests that this is a bacterial strain that was obtained before, but was it truly so? Were the bacteria identified using any molecular methodology? Did the strain originate from a known library, was it commercially purchased? Quite obviously, many questions need to be answered here, which means that this paragraph needs to be expanded.
Line 121: Which species were males and which females?
Line 126: Correct to "approximately".
Lines 130-131: Correct all three instances of "was" to plural "were".
Line 136: Delete "and the fish was fed twice daily", it was already stated in Line 132.
Line 137: Delete "period".
Line 139: How often was the water exchanged by 50%?
Line 140: Correct to singular "six-week" and to plural "groupers". Also change "for" to "were".
Line 141: Change "was" to plural "were".
Line 145: Delete this whole sentence, keep only the equation.
Line 170: Change "was" to plural "were".
Line 171: Change to plural "groupers".
Line 175: Add "count" after "cell".
Line 187: Change "record" to "second recording of".
Line 192: Change to singular "Leukocyte".
Lines 196-198: Delete this whole parenthesis apart from "(PBS)" - PBS is such an universal solution that it does not require to be introduced in such detail.
Lines 206-207: Change to "grouper spleens were".
Line 221: Change to plural "beads ... were".
Results: Apart from linguistic mistakes, I have spotted significant issues regarding the statistical analysis of data, all of which requires crucial changes in data preparation and processing to be introduced.
Lines 234-235: Correct to "dietary oleic acid supplementation".
Lines 236-237: Delete both "ratios".
Line 237: Delete "the group of".
Line 238: Replace "samples" with "parameters".
Line 239: Delete "also".
Line 242: Change word order to "dietary fatty acids".
Figure 2: I suggest to set this picture in a panel with a non-challenged fish.
Line 257: Change to singular "count".
Table 3: How were these percentages for means and SEM obtained for the survival, if there were 3x10 fish in every group? I suggest to indicate also the raw number of survived fish as "x/30", for clarity. Also, what is the n for the RBC and WBC counts? If it is 3, then this is the same issue as mentioned below for Table 4.
Line 268: Change word order to "dietary oleic acid formulation". Also, delete the second "dietary".
Line 270: Change to plural "formulations".
Line 271: Delete "dietaries".
Line 272: Insert "was" infront of "detected".
Line 273: Change to plural "formulations". Also, delete "dietaries".
Line 280: Correct "can" to "could".
Line 282: Correct to "diets".
Line 286: Correct "supplemented dietary" to "supplementation".
Line 287: Delete the parenthesis around "49.6%".
Line 291: Correct to "diet".
Line 298: Correct to singular "acid".
Table 4: This table repeats the same information already presented in Figure 3. The only new thing is the indication of significant differences for each group pre- and post-challenge, but such information could easily be shown also in Figure 3. Thus, I advise to add these capital letters in Figure 3 and to remove Table 4 completely. However, this brings us to a more important, data-quality related issue - why is all data presented as averages of n=3? There were 3 fish/tank sampled, which gives n=9. In addition, every sample was analyzed in triplicate, leaving the actual n at 27. Of course, this way the SEM will be larger, but that is also the proper way of presenting and calculating data for statistics, if the Authors want to use a parametric test such as ANOVA. Meanwhile, n=3 will never show a normal distribution of data, thus it would be required to use the non-parametric Kruskal-Wallis test, not ANOVA.
Discussion and Conclusions: Apart from necessary linguistic corrections, this content of these two sections can not be assessed completely due to the aforementioned problems with statistical analysis, which the discussion is based on. The Authors did a decent job in the argumentation, but as mentioned before, some of these conclusions might change depending on the actual statistical significance of differences between groups.
Lines 315-316: Delete "powerful tool" and "approach".
Line 317: Replace "to assess" with "about", and "contributes to a better understanding of the" with "and its contribution in the".
Line 320: Insert "effects" infront "improving".
Line 322: Correct "daily dietary" to "diets".
Line 323: Correct "The result" to "This".
Line 326: Correct to singular "eicosanoid".
Line 328: Delete "than the control group".
Line 329: Replace "indicating the composition of" with "in which".
Line 332: Correct to "challenge" and "consist".
Line 336: Replace "it provides" with "they are", and "to identify and evaluate" with "of".
Line 338: Delete "haematological parameter represents the fish stress level, where".
Line 339: Insert "was" infront "attributable". Also, correct to "peaking".
Line 340: Delete "of".
Line 341: Delete "However, as referred to the WBC count (Table 3),".
Line 342: Replace "diets indicated" with "groups indicating that".
Line 343: Delete ", and not causing intolerance".
Line 344: Delete "components for".
Line 345: Delete "body tissues including structural". Also, I do not even know what "extensive metabolic" was supposed to mention.
Lines 347-348: Improve the style: "has been positively associated with the increase in the survival rate of grouper in the current study,".
Line 351: Change "the" to "this".
Line 354: Correct "are" to past "were".
Line 355: Correct "Their findings" to "They".
Line 362: Delete "fatty acid known as".
Line 367: Correct "dietaries" to "dietary supplements".
Line 370: Correct to singular "metabolism".
Line 372: Correct "of other ARA fatty acid dietary" to "with ARA".
Lines 378-379: Delete this whole first sentence, it is of no use. Also, delete "that forms part".
Line 384: Delete "dietary".
Lines 384-385: "where lysozyme is the main effector molecule that involves in anti-bacterial and anti-inflammatory activities during infections" - this sentence should be moved and merged with the first sentence of this paragraph, it does not make sense in its current form.
Line 387: Delete "study".
Line 388: Change word order to "dietary oleic acid". Also, change to "comparing".
Line 404: Correct to "principle", insert "in" infront "which" and change "enhanced" to "increase".
Line 405: Delete "lysozyme activity as phagocytes".
Line 406: Delete "in these cells".
Lines 407-408: Delete the whole first half of this sentence.
Lines 410-411: Correct to "reports". Also, delete "can be applied for health beneficial as oleic acid" and "that involve".
Line 413: Insert "that" infront of "the survival".
Line 415: Change "composition" to "content".
Lines 416-417: Delete "the finding in" and "of oleic acid as anti-inflammatory".
Line 420: Delete "dietary".
Line 421: Correct to "increment".
Lines 423-425: I struggle to understand what this sentence is supposed to indicate.
Line 424: Delete "dietary".
Line 426: Replace "is found to have" with "has".
Line 429: Replace "composition" with "inclusion".
Line 430: Delete "that can be used".
Line 431: Correct to plural "juveniles".
Line 434: Delete "the".
Line 436: Delete "infection".
Line 441: Delete "in the zebrafish".
Line 455: Correct to singular "challenge". Also, change word order to "dietary oleic acid".
Line 457: Delete "compound".
Line 458: Correct to "economic".

Reviewer 3 Report
Introduction:
1- The authors must emphasize on the negative effects of antibiotics on host. Antibiotics are found potentially immunosuppresive, oxidative, and fail to protect against multi-pathogenic diseases. See the following references, for example:
Aquaculture Research, 51(10), 4215-4224.
Aquaculture nutrition, 25(2), 298-309.
Veterinary immunology and immunopathology, 67(4), 317-325.
Aquaculture Reports, 19, 100602.
2- The authors must focus on literature regarding using relevant fatty acids (those used in this study), rather than essential oils, EPA, DHA ...
Methods:
1- Why the authors referred to previous studies (line 99) when there are modifications?! Why the authors replaced fatty acids with cellulose not the vegetable oil?! This make a great difference in dietary fat and energy among the treatments.
2- Line 106: extrusion is different to what the authors did. It is pelleting not extrusion.
3- Diet composition and fatty acid profile are needed.
4- Fig. 1 is not necessary
5- Line 139: 50% per day? What about water quality parameters?
6- Challenge period is not clear. Line 156 tells it is 30 min immersion challenge, but line 164 tells it is one week challenge.
7- sampling is not clear. There were 15 fish per tank, the authors sampled three fish before challenge, then they challenged 10 fish of the remaining 12 fish. These challenged (10) and non-challegned (2) fish were kept in a same aquarium. How the authors sampled the fish after challenge?!
8- Statistical analysis needs revision. The authors must check normal distribution and variance homogeneity before ANOVA. Moreover, the authors must use two way ANOVA, as there were two factors, i.e. dietary fatty acids and bacterial challenge.
Results:
this section will be changed after revising statistics.
the authors must avoid presenting a single result in both figure and table.
exact p-value and sample size must be added to the results.
Reviewer 4 Report
The authors investigated the effect of different fatty acids dietary on the immune response of hybrid grouper (Epinephelus fuscoguttatus x Epinephelus lanceolatus) challenged with Vibrio vulnificus. They designed five treatments to test their hypothesis. This manuscript (MS) was clearly written and easy to understand. Although they covered a wide range of parameters, some gaps are still in provided data. This work could help the sustainability of this species farming if more data could be provided with a longer experiment design. Some major issues significantly compromised the quality of this MS.
Major comments:
- First, the manuscript needs to be edited by a native English speaker to improve the language of the MS and fix errors.
- They did not provide adequate data and evidence to support their hypothesis. Publishing in this journal is competitive and more data should be provided.
- The experimental period is not long enough to make a solid conclusion regarding data. The result of this experiment, especially growth data is not reliable enough.
- The formulation of diets has fundamental problems such as including too much soybean and not enough fish meal and fish oil.
- This work has not had enough novelty as well.
- Due to these issues, I did not go through line-by-line comments.
Kind regards
Round 2
Reviewer 1 Report
Thank you for addressing my comments. The manuscript is now acceptable for publication.
Author Response
Thank you for your response.
Reviewer 2 Report
The Authors of the revised manuscript titled "Comparison of different fatty acids dietary on the immune response of hybrid grouper (Epinephelus fuscoguttatus x Epinephelus lanceolatus) challenged with Vibrio vulnificus" have put forth a significant amount of work in order to introduce all the necessary changes which their paper required, and they succeeded in achieving that goal. I am especiially fond of the reworked statistical analysis of data, it definitely improved the merit of this manuscript. There are still some minor linguistic problems lingering here and they, but I believe this will be fixed by the editors during final draft preparation. All in all, good work.
Author Response
Thank you for your comments and suggestions.
Reviewer 3 Report
.
Author Response
Thank you for your response.
Reviewer 4 Report
The authors have not improved the quality of the MS and my decision is still "reject"
Kind regards
Author Response
Thank you for your comments.